# The Association of Bisphenol A and Phthalates with Risk of Breast Cancer: A Meta-Analysis

**DOI:** 10.3390/ijerph18052375

**Published:** 2021-03-01

**Authors:** Ge Liu, Wei Cai, Huan Liu, Haihong Jiang, Yongyi Bi, Hong Wang

**Affiliations:** 1Department of Occupational and Environmental Health, School of Health Sciences, Wuhan University, Wuhan 430071, China; liuqiuge876@163.com (G.L.); 18844504051@163.com (W.C.); lhlmlzw@126.com (H.L.); 15735010286@163.com (H.J.); yongyib@aliyun.com (Y.B.); 2Hubei Biomass-Resource Chemistry and Environment Biotechnology Key Laboratory, Wuhan University, Wuhan 430071, China

**Keywords:** bisphenol A, phthalates, endocrine-disrupting chemicals, breast cancer, meta-analysis

## Abstract

Background: Breast cancer is the most common cancer and the second leading cause of cancer-related death amongst American women. Endocrine-disrupting chemicals (EDCs), especially bisphenol A (BPA) and phthalates, have adverse effects on human health. However, the association of BPA and phthalates with breast cancer remains conflicting. This study aims to investigate the association of BPA and phthalates with breast cancer. Methods: Correlative studies were identified by systematically searching three electronic databases, namely, PubMed, Web of Sciences, and Embase, up to November 2020. All data were analyzed using Stata 15.0. Results: A total of nine studies, consisting of 7820 breast cancer cases and controls, were included. The urinary phthalate metabolite mono-benzyl phthalate (MBzP) and mono-2-isobutyl phthalate (MiBP) were negatively associated with breast cancer (OR = 0.73, 95% CI: 0.60–0.90; OR = 0.75, 95% CI: 0.58–0.98, respectively). However, the overall ORs for BPA, mono-ethyl phthalate (MEP), mono-(2-ethyl-5-hydroxyhexyl) phthalate (MEHHP), mono-2-ethylhexyl phthalate (MEHP), mono-(2-ethyl-5-oxohexyl) phthalate (MEOHP), mono-(3-carboxypropyl) phthalate (MCPP), and mono-butyl phthalate (MBP) were 0.85 (95% CI: 0.69–1.05), 0.96 (95% CI: 0.62–1.48), 1.12 (95% CI: 0.88–1.42), 1.13 (95% CI: 0.74–1.73), 1.01 (95% CI: 0.74–1.40), 0.74 (95% CI: 0.48–1.14), and 0.80 (95% CI: 0.55–1.15), respectively, suggesting no significant association. The sensitivity analysis indicated that the results were relatively stable. Conclusion: Phthalate metabolites MBzP and MiBP were passively associated with breast cancer, whereas no associations were found between BPA, MEP, MEHHP, MEHP, MEOHP, MCPP, and MBP and breast cancer. More high-quality case-control studies or persuasive cohort studies are urgently needed to draw the best conclusions.

## 1. Introduction

Endocrine-disrupting chemicals (EDCs) are ubiquitous exogenous chemical substances that interfere with a series of functions of endocrine system either by enzyme and receptor-mediated mechanisms or epigenetic effects, thereby adversely inducing various aspects of reproductive, metabolic, neurological, and immune problems at any stage of human life [1,2]. Phthalates and bisphenol A (BPA) are two prevailing EDCs that have drawn increasingly remarkable public concern globally over the past few decades.

Phthalates are a group of aromatic chemicals consisting of dialkyl esters or alkyl and aryl esters of orthophthalic acid (1,2-dicarboxylic acid) [3], and can be commonly separated into two categories, i.e., high molecular weight (HMW) phthalates and low molecular weight (LMW) phthalates [4]. HMW phthalates, such as di-(2-ethylhexyl) phthalate (DEHP), di-n-octyl phthalate (DOP), di-isononyl phthalate (DiNP), and butyl benzyl phthalate (BBzP), mainly function as plasticizers in plastics products, food containers, children’s toys, building materials, and medical devices (e.g., medical tubing, blood and intravenous bags, dialysis machines, and disposable surgical gloves) [5,6,7]. LMW phthalates, such as di-butyl phthalate (DBP), di-isobutyl phthalates (DiBP), di-ethyl phthalate (DEP), and di-methyl phthalate (DMP), are widely used as additives in personal care products (e.g., cosmetics, lotions, and perfumes), medications and dietary supplements [8,9] and pesticide [10]. In this way, it is important to notice that due to the fact that phthalates are usually physically, rather than chemically, added to the above daily products. The ease release of phthalates into the environment results in pervasive human exposure to environmental phthalates through digestive tract ingestion (dietary intake or indoor dust ingestion) [11,12], inhalation [13,14], and percutaneous absorption (by personal care products) directly or indirectly. The phenomenon occurs because phthalates are usually physically, rather than chemically, added to the above daily products.

BPA is an alkylphenol organic compound whose structure consists of two phenol functional groups connected by two carbon-containing methyl groups [15]. BPA is extensively used in epoxy resins and polycarbonate plastics, and shares some common sources of exposure with phthalates. A range of consumer products that contain BPA can release hazardous pollutants under high temperature, strong acid, or other certain conditions. For example, a previous study has tested that the migration of BPA from polycarbonate plastics into the food occurred at 70 °C [16], and this occurrence is consequently considered as a potential chronic threat to human health [17,18].

In fact, a number of EDCs have been increasingly reported to be linked with human health problems especially female reproductive disorders in recent years [19]. Results from several literature indicated that BPA and phthalates played negative roles in thyroid function on the basis of different thyroid hormone levels [20,21]. In addition, two cross-sectional studies have found positive relationship between BPA and the phthalate metabolite mono-ethyl phthalate (MEP) and precocious puberty in Thai girls [22,23]. Some other epidemiologic studies have focused on the higher BPA level in women with polycystic ovarian syndrome (PCOS) [24,25] and the consistent conclusion can be also found in adolescents [26]. In addition, various cell cultures [27], animal models [28], and human epidemiologic data have supplied evidences that phthalates may change the endometrial environment or are possibly associated with endometriosis [29]. One recent study has described that BPA exposure may affect the expression of key severe acute respiratory syndrome coronavirus 2 (SARS-CoV-2) infection mediators such as angiotensin-converting enzyme 2 (ACE2), transmembrane serine protease 2 (TMPRSS2), and furin in human tissues. Thus, the BPA exposure may have relevant effects on the risk of severe COVID-19 [30]. Furthermore, BPA and phthalates—as oestrogen-mimicking EDCs—are thought to be responsible for inducing the occurrence and progression of many oestrogen-dependent cancers by binding to oestrogenic receptors, which are important evolution-related transcription factors related to many critical biological processes. Such actions probably play a considerable role in human breast carcinogenesis [19,31]. Amongst the multitudinous hormone-dependent cancers related to BPA and phthalates exposure, breast cancer is the most important issue that should be placed more attention.

Breast cancer is the most frequent cancer and the second leading cause of cancer death amongst American women [32] and accounts for almost a quarter of all types of cancer in women worldwide. The incidence of breast cancer is on the rise in many countries over the last few decades [33]. Over time, emerging literature from laboratory works and human epidemiological studies have focused on the relationship of BPA and phthalates exposure with breast cancer [34,35,36]. However, the results still remain inconsistent although several recent laboratory findings have supplied valuable evidence that BPA or di(2-ethylhexyl) phthalate (DEHP) exposure is responsible for the development and progression of breast cancer [17,37]. An Alaska case-control study has revealed that the phthalate metabolite MEHP was associated with breast cancer [38]. Similarly, another study has demonstrated the correlation between urinary MEP concentrations and increased breast cancer risk, whereas MBzP and mono(3-carboxypropyl) phthalate (MCPP) were negatively linked with breast cancer [39]. However, in a recent study conducted in America, the researchers have not observed any relationship between phthalates or BPA and breast cancer [40].

Therefore, given the controversial conclusion of current epidemiological studies, we aim to perform a meta-analysis to assess systematically the association of BPA and eight phthalate metabolites, including mono-benzyl phthalate (MBzP), mono-ethyl phthalate (MEP), mono-(2-ethyl-5-hydroxyhexyl) phthalate (MEHHP), mono-2-ethylhexyl phthalate (MEHP), mono-(2-ethyl-5-oxohexyl) phthalate (MEOHP), mono (3-carboxypropyl) phthalate (MCPP), mono-butyl phthalate (MBP), and mono-2-isobutyl phthalate (MiBP) with the risk of breast cancer.

## 2. Methods

### 2.1. Literature Search Strategy

We performed a systematic literature search in three online databases, namely, PubMed, Web of Sciences, and Embase for eligible studies, with the following terms from January 1990 to November 2020: (1) “bisphenol A” OR “BPA” OR “phthalates” OR “phthalate” OR “phthalate esters” OR “phthalic acid ester” OR “endocrine-disruptors” OR “endocrine-disrupting chemicals”; (2) “breast cancer” OR “mammary cancer”; and (3) 2 and 3. First, we browsed the titles and abstracts to exclude irrelevant literature (such as cell experiments, animal experiments, the studies aimed on the association of other EDCs exposure and other health outcomes), and potential eligible articles were identified by reading the full text carefully. All retrieved results were exported into the NoteExpress to remove duplicates. The references cited in the included studies and the references of other relevant reviews and meta-analyses were also searched manually so that additional eligible sources would not be neglected.

### 2.2. Inclusion and Exclusion Criteria

The inclusion criteria of our meta-analysis were as follows: cohort or case-control studies (1) that evaluated the relationship between BPA or phthalates exposure and breast cancer; (2) that odds ratio (OR) or relative risk (RR) and their 95% confidence intervals (95% CI) were reported; and (3) published in English. Letters, reports, reviews and editorials, conference abstracts, and cross-sectional studies were excluded. We also excluded studies reporting only urinary concentrations of EDCs but without available OR or RR information. Overall, three investigators (Ge Liu, Huan Liu, and Haihong Jiang) screened and assessed the eligibility for all articles independently, and any disagreement was discussed and resolved by consensus.

### 2.3. Data Extraction and Quality Assessment

The extraction and recording of data and quality assessment were carried out by three independent investigators (Ge Liu, Huan Liu, and Haihong Jiang), and any debate was settled by discussion. For each selected study, the following items were extracted: the first author and the year of publication, study area, study period, the type of study, number of cases and controls, age, and interested categories of EDCs. The quality appraisal of all the articles included was performed on the basis of Newcastle–Ottawa Scale (NOS) with three categories including selection, comparability, and exposure for case control studies, or outcome for cohort studies [41]. A high NOS score resulted in a high study quality. Studies with scores of no less than seven were considered to be of high quality.

### 2.4. Statistical Analysis

All extracted ORs were calculations from the highest versus the lowest concentrations of interested EDCs. Statistical analysis was carried out using Stata version 15.0 software. (Stata Corp, College Station, TX, USA) to coalesce the summary ORs and relevant 95% CIs of the relationship of BPA and phthalates exposure with breast cancer. The size of heterogeneity amongst the included studies was measured using Chi-square test and I^2^ statistic. If I^2^ < 50% and *p* > 0.05, no statistical heterogeneity was present, and the fixed-effects model was used. Otherwise, the random-effects model was selected for a merger. The subgroup analysis was conducted to assess the influence of study region and source of controls on the estimation of overall OR. We also conducted leave-one-out sensitivity analysis to explore the potential sources of heterogeneity and meanwhile to assess the robustness of the results. Considering that fewer than 10 eligible studies were included in our meta-analysis, no funnel plot test for publication bias was performed [42].

## 3. Results

### 3.1. Literature Search and Selection

Figure 1 shows the flow diagram of the literature screening process. A total of 2940 potentially relevant studies were presented in accordance with the key words preliminarily searched in three databases. Following the deletion of duplicate records, 2388 articles remained. Of these articles, 2077 unrelated studies were excluded by scanning all titles and abstracts. A total of 311 full-text articles were carefully read and assessed for data eligibility, and 302 studies were removed on account of conference abstract, review, and no available data. Eventually, the remaining nine articles were included in the final meta-analysis [38,39,40,43,44,45,46,47,48]. Amongst these articles, five focused on BPA, whereas six paid attention to phthalates, and two studies investigated both of the EDCs.

### 3.2. Characteristics of Included Studies

In total, nine retrospective case-control studies, involving 7820 participants, were included. The maximum sample size of participants was 2501, and the minimum was only 50. Overall, six studies were conducted on the basis of participants in America, and the remaining works were performed in Alaska Native, Poland, and Northern Mexico. Table 1 displayed the detailed characteristics of all enrolled studies. The results of quality assessment by NOS for the included studies were shown in Appendix A. Of the 9 studies, 5 studies were deemed to have high quality (scored 7–9), whereas other studies had medium quality (scored 4–6).

### 3.3. BPA Levels and Risk of Breast Cancer

Overall, five studies including 5668 participants (1661 breast cancer cases and 4007 controls) were conducted to evaluate the association between BPA and the risk of breast cancer. We selected the fixed-effects model in pooled analysis because no heterogeneity (I^2^ = 0.0%, *p* = 0.705) was found amongst these studies. As shown in Figure 2, the overall OR for the highest versus lowest BPA levels and breast cancer risk was 0.85 (95% CI: 0.69–1.05), indicating no significant association.

### 3.4. Urinary Phthalate Metabolite Levels and Risk of Breast Cancer

We, respectively, analyzed the association between eight different phthalate metabolites and breast cancer risk. Figure 3a showed the lack of heterogeneity in six included studies regarding MBzP (I^2^ = 27.0%, *p* = 0.232). Thus, the fixed-effects model was used. Comparing the highest versus the lowest levels, the urinary MBzP was negatively associated with the risk of breast cancer (OR = 0.73, 95% CI: 0.60–0.90). In addition, the OR for urinary phthalate metabolite MiBP and risk of breast cancer was 0.75 (95% CI: 0.58–0.98, Figure 3h) with no heterogeneity (I^2^ = 0.0%, *p* = 0.937), which also showed a negative correlation. However, the overall ORs for MEP, MEHHP, MEHP, MEOHP, MCPP, and MBP, were 0.96 (95% CI: 0.62–1.48, Figure 3b), 1.12 (95% CI: 0.88–1.42, Figure 3c), 1.13 (95% CI: 0.74–1.73, Figure 3d), 1.01 (95% CI: 0.74–1.40, Figure 3e), 0.74 (95% CI: 0.48–1.14, Figure 3f), and 0.80 (95% CI: 0.55–1.15, Figure 3g), respectively, showing the six aforementioned phthalate metabolites had no significant association with breast cancer. In addition, results revealed no heterogeneity between the articles assessing the relationship of MEHHP (I^2^ = 0.0%, *p* = 0.497), MBP (I^2^ = 0.0%, *p* = 0.843) and MEOHP (I^2^ = 0.0%, *p* = 0.587) with breast cancer, but significant heterogeneity was found amongst articles on MEP (I^2^ = 73.2%, *p* = 0.005) MEHP (I^2^ = 59.7%, *p* = 0.059), and MCPP (I^2^ = 80.0%, *p* = 0.007).

### 3.5. Sensitivity Analyses

Regarding the sensitivity analysis, the removal of the study of L. López-Carrillo (2010) [39] study reduced I^2^ from 73.2 to 0.9% and increased *p* from 0.001 to 0.388 in the correlation analysis of urine phthalate metabolite MEP and breast cancer risk, whereas the removal of the study of A K. Holmes (2014) [38] study reduced I^2^ from 59.7% to 1.0% and increase *p* from 0.059 to 0.364 for the association of urine MEHP and breast cancer. These results indicated the potential sources of heterogeneity. The correspondingly total ORs value for MEP and MEHP did not change significantly, suggesting that the results of the meta-analysis were relatively stable. However, when we removed the study of K W. Reeves (2019) [47] study, reduced I^2^ from 83.8 to 43.9%, increased *p* from 0.002 to 0.182 for the association of urine MCPP and breast cancer, and decreased OR from 0.80 (95% CI: 0.41–1.56) to 0.60 (95% CI: 0.38–0.94). Detailed results of sensitivity analyses were displayed in Appendix A.

### 3.6. Subgroup Analyses

We performed subgroup analyses according to study region and source of controls for BPA, MBzP, MEP, MEHHP, MEHP, MEOHP, MCPP, and MBP. Specially, we performed the subgroup analysis of the relationship between MiBP and breast cancer only according to the study region since the control sources were all from the general population. The results were displayed successively in Table 2, Table 3, Table 4, Table 5, Table 6, Table 7, Table 8, Table 9 and Table 10.

As shown in Table 2, BPA was significantly adversely associated with breast cancer risk for studies conducted in America (OR = 0.78, 95% CI: 0.61–0.99, I^2^ = 0.0%), but no specific association was observed in the subgroup of the source of controls.

As for phthalate metabolites, MBzP was significantly adverse associated with breast cancer in the subgroups of America (OR = 0.74, 95% CI: 0.59–0.93, I^2^ = 0. 0%) and the general population controls (OR = 0.66, 95% CI: 0.51–0.85, I^2^ = 0.0%), but no association was observed in either non-America or clinical medical center subgroups (Table 3). Interestingly, as shown in Table 8, we also observed that MCPP was passively associated with breast cancer in Non-America (OR = 0.44, 95% CI: 0.24–0.80) and the general population control subgroups (OR = 0.66, 95% CI: 0.51–0.85, I^2^ = 43.9%). Moreover, MEP, MEHHP, MEHP, MEOHP, MBP, and MiBP were not significantly associated with breast cancer risk in the subgroups (Table 4, Table 5, Table 6 and Table 7, Table 9, and Table 10).

## 4. Discussion

The widespread sources of exposure to BPA and phthalates are unavoidable and unintentional in daily life. Moreover, BPA and phthalates all exist as immanent EDCs that play a major role in adverse reproductive system damages [49,50,51], metabolic disorders [52], grip strength [53], and hormone-sensitive cancers [54,55] in males and females. Previous studies have shown that exposure during preconception and prenatal stages may cause adverse health consequences for pregnant women and their offspring [18,56,57,58]. An earlier review includes examples that showing how several EDCs are identified and affect the development of mammary glands in animal models [59]. Fabiana and Kalpana’s recent studies conducted in rat models report that exposure to low-dose diethyl phthalate (DEP) and phenol at critical windows lead to changes in histology and transcriptome in rat mammary glands [60,61]. However, very few epidemiological studies are conducted on the correlation between certain EDCs, especially BPA and phthalates and human breast cancer risk. Some studies have even yielded inconsistent results.

Therefore, we conducted an updated meta-analysis by extracting and summarizing the data from nine selected case-control studies, which involves 7820 participants across four different countries, to assess the relationship of BPA and eight phthalate metabolites with the risk of breast cancer accurately and comprehensively. We found that only phthalate metabolites MBzP and MiBP were passively associated with breast cancer risk. One of the possible mechanisms leading to the negative relationship is the ability of MBzP to activate human peroxisome proliferator-activated receptor (PPAR) α and γ [62], and the ligand activation of PPARγ is relative to adipocyte differentiation, lipid accumulation, and decreased growth of breast cancer cell [63,64]. Another potential speculation is that phthalates metabolites are related to increased intron 1 methylation level and upregulated disintegrin and metalloproteinase domain 33 (ADAM33) expression, which play an important role in reducing breast cancer risk [35]. Besides, results did not provide sufficient evidence of the correlations of BPA and other phthalate metabolites MEP, MEHHP, MEHP, MEOHP, MCPP, and MBP with breast cancer. However, when we conducted subgroup analyses, a marginally negative association between BPA exposure and breast cancer risk was observed in America but not in Non-America; for MCPP, we found a negative association in Non-America and general population subgroups and for MBzP, the passive association between urinary MBzP and breast cancer remained in subgroups of America and general population controls. These results proved that the correlation was influenced by region and control selection, which might partly contribute to different sources or levels of BPA and phthalates exposure in different countries and control populations. The subgroup analyses for MEP, MEHHP, MEHP, MBP, and MEOHP did not reveal any significant result. Additionally, the results of sensitivity analysis suggested that the results of our meta-analysis were relatively stable, except that the conclusion was reversed after the removal of the study of K W. Reeves (2019) [47] study in the correlation analysis for MCPP and breast cancer. One possible explanation was that the subjects of the study were recruited from postmenopausal women, which was different from two other studies. Finally, regional diversity, age differences, different detection methods, the limit of detection (LOD) and concentration classification methods for EDCs, and adjusted models produced heterogeneity to some extent and affected the data merging.

Our study had some advantages. Firstly, this study was the first meta-analysis to evaluate systematically the relationship of exposure to BPA and eight specific phthalate metabolites with breast cancer risk across four different countries to date. Secondly, we also discovered the potential sources of heterogeneity according to leaving a study each time in the sensitivity analysis. Finally, we performed the subgroup analyses by region (i.e., America and Non-America) and sources of control (i.e., clinical medical center and general population), which further verified the relationship between selected EDCs and breast cancer risk.

Our research also had several limitations. Firstly, the meta-analysis was based on relatively insufficient number of studies so that only one study was included in a certain subgroup analysis. Secondly, the sensitivity analysis suggested the presence of heterogeneity, and some of the results were not stable. In this sense, well-conducted epidemiological studies on the relationship of BPA and various phthalate metabolites with breast cancer are needed to reinforce our research results. Thirdly, although urine concentrations of BPA and phthalate metabolites are the most common methods to assess human BPA and phthalates exposures, some potential factors such as the relatively short biological half-life of above EDCs, the changes of subjects’ lifestyle after breast cancer diagnosis, the nature of retrospective case-control study, breast cancer itself, and the treatments or surgery are all likely to affect the measurement of EDCs exposure. Therefore, the time of urine collection, the inconsistency of the assessment and classification of exposure levels of the same EDCs in different original articles and the evaluation of long-term exposure based on short-term biomarker levels measured in urine led to a degree of bias. More sensitive and accurate biomarkers and better designed studies are needed to explore this possible effect in the future.

## 5. Conclusions

The meta-analysis results demonstrated that phthalate metabolites MBzP and MiBP were passively associated with breast cancer risk. However, MEP, MEHHP, MEHP, MEOHP, MCPP, MBP, and BPA were not statistically associated with breast cancer. Given the limited epidemiological data on these EDCs and risk of breast cancer, high-quality case-control studies or persuasive cohort studies are urgently needed to draw the best conclusions.

## Figures and Tables

**Figure 1 ijerph-18-02375-f001:**
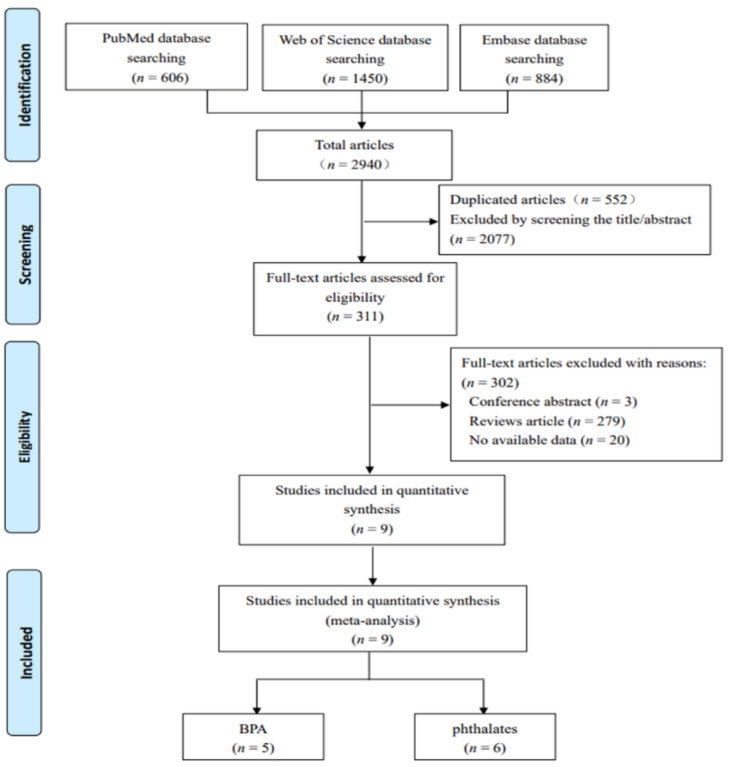
The flow diagram of screening process of included studies.

**Figure 2 ijerph-18-02375-f002:**
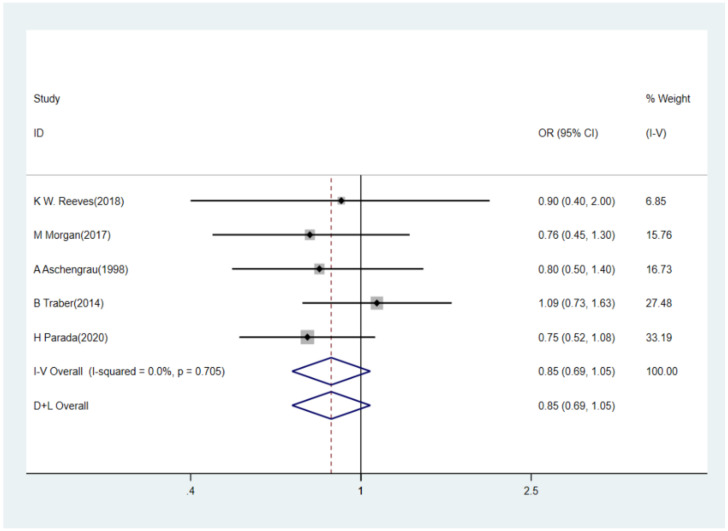
Forest plot of pooled OR of BPA level and breast cancer risk. Abbreviations: OR, odds ratio; BPA, bisphenol A.

**Figure 3 ijerph-18-02375-f003:**
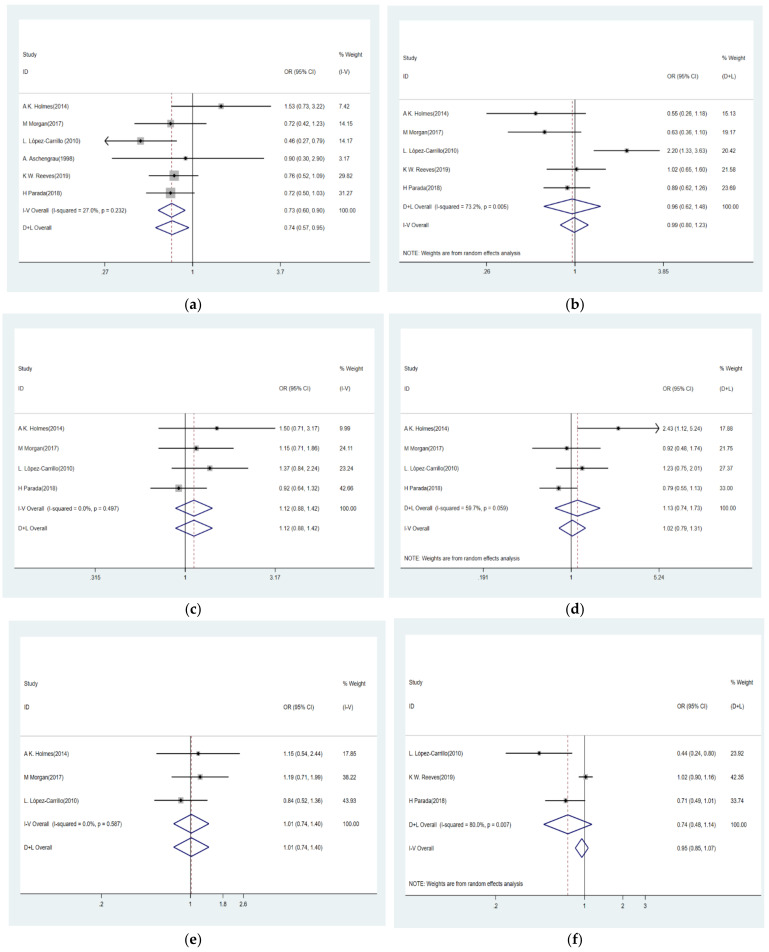
Forest plot of pooled OR of phthalate metabolites and breast cancer risk: (**a**) MBzP; (**b**) MEP; (**c**) MEHHP; (**d**) MEHP; (**e**) MEOHP; (**f**) MCPP; (**g**) MBP; (**h**) MiBP. Abbreviations: MBzP, mono-benzyl phthalate; MEP, mono-ethyl phthalate; MEHHP, mono-(2-ethyl-5-hydroxyhexyl) phthalate; MEHP, mono-2-ethylhexyl phthalate; MEOHP, mono-(2-ethyl-5-oxohexyl) phthalate; MCPP, mono (3-carboxypropyl) phthalate; MBP, mono-butyl phthalate; MiBP, mono-2-isobutyl phthalate.

**Table 1 ijerph-18-02375-t001:** Characteristics of the included studies.

Study (Year)	Region	Study Design	Time Period	No. of Case/Control	Age	Categories of EDCs	Samples Determined
A K. Holmes (2014) [38]	Alaska Native	case-control	1999–2002	75/95	30–88	MBzP, MEP, MBP, MEHHP, MEHP, MEOHP, MMP	Urine
L. López-Carrillo (2010) [39]	Northern Mexico	case-control	2007–2008	233/221	Cases: 53.41 ± 12.78 Controls:53.83 ± 12.54	MBzP, MEP, MCPP, MBP, MEHHP, MEHP, MEOHP, MiBP, MECPP	Urine
M Morgan (2017) [40]	America	case-control	2003–2010	91/2410	≥20	MBzP/MZP, MEP, MBP, MEHHP, MEHP, MEOHP, MiBP, MnBP, MCCP, MCPP	Urine
M Morgan (2017) [40]	America	case-control	2005–2010	78/2067	≥20	BPA	Urine
K W. Reeves (2018) [43]	America	case-control	2014–2015	36/14	Case: 55.7 ± 10.5 Control: 42.1 ± 16.5	BPA	Breast adipose tissue
A. Aschengrau (1998) [44]	America	case-control	1983–1986	261/753	≥18	BBzP	Urine
A Aschengrau (1998) [44]	America	case-control	1983–1986	261/753	≥18	BPA	NIOSH/NOES
B Traber (2014) [45]	Poland	case-control	2000–2003	575/575	20–74	BPA-glucuronide (BPA-G)	Urine
H Parada (2019) [46]	America	case-control	1996–1997	711/598	22–96	BPA	Urine
K W. Reeves (2019) [47]	America	case-control	NA	404/768	Case: 62.56; Control: 62.46	BzP, MEP, MCPP, DBP, DiBP, MCNP, MCOP, DEHP	Urine
H Parada (2018) [48]	America	case-control	1996–1997	710/598	22–96	MBzP, MEP, MCPP, MEHHP, MEHP, MEOHP, MiBP, MnBP, MECPP, MCOP, MCNP	Urine

Abbrevations: NA, not available; NIOSH/NOES: National Institute for Occupational Safety and Health National Occupational Exposure Survey.

**Table 2 ijerph-18-02375-t002:** Subgroup analysis of the association between BPA and breast cancer.

Scheme	No. of Studies	Meta-Analyses		Model	Heterogeneity	
		ORs (95% CIs)	*p*-Value		I^2^	*p*-Value
Region						
Non-America	1	1.09 (0.73–1.63)	0.674	-	-	-
America	4	0.78 (0.61–0.99)	0.045	Fixed	0.0	0.980
Source of controls						
Clinical medical centre	1	0.90 (0.40–2.01)	0.797	-	-	-
General population	4	0.85 (0.68–1.06)	0.142	Fixed	0.0	0.542

**Table 3 ijerph-18-02375-t003:** Subgroup analysis of the association between MBzP and breast cancer.

Subgroups	No. of Studies	Meta-Analyses		Model	Heterogeneity	
		ORs (95% CIs)	*p*-Value		I^2^	*p*-Value
Region						
Non-America	2	0.82 (0.25–2.64)	0.734	Random	84.9	0.010
America	4	0.74 (0.59–0.93)	0.010	Fixed	0.0	0.983
Source of controls						
Clinical medical center	2	1.00 (0.51–1.95)	0.996	Random	63.4	0.098
General population	4	0.66 (0.51–0.85)	0.001	Fixed	0.0	0.503

**Table 4 ijerph-18-02375-t004:** Subgroup analysis of the association between MEP and breast cancer.

Subgroups	No. of Studies	Meta-Analyses		Model	Heterogeneity	
		ORs (95% CIs)	*p*-Value		I^2^	*p*-Value
Region						
Non-America	2	1.13 (0.29–4.40)	0.856	Random	88.8	0.003
America	3	0.87 (0.68–1.11)	0.259	Fixed	0.0	0.411
Source of controls						
Clinical medical center	2	0.87 (0.59–1.28)	0.480	Fixed	47.1	0.169
General population	3	1.07 (0.55–2.10)	0.836	Random	83.9	0.002

**Table 5 ijerph-18-02375-t005:** Subgroup analysis of the association between MEHHP and breast cancer.

Subgroups	No. of Studies	Meta-Analyses		Model	Heterogeneity	
		ORs (95% CIs)	*p*-Value		I^2^	*p*-Value
Region						
Non-America	2	1.41(0.93–2.12)	0.102	Fixed	0.0	0.843
America	2	1.00 (0.75–1.33)	0.985	Fixed	0.0	0.468
Source of controls						
Clinical medical center	1	1.50 (0.71–3.17)	0.288	-	-	-
General population	3	1.08 (0.84–1.39)	0.533	Fixed	0.0	0.423

**Table 6 ijerph-18-02375-t006:** Subgroup analysis of the association between MEHP and breast cancer.

Subgroups	No. of Studies	Meta-Analyses		Model	Heterogeneity	
		ORs (95% CIs)	*p*-Value		I^2^	*p*-Value
Region						
Non-America	2	1.62 (0.84–3.11)	0.150	Random	52.9	0.145
America	2	0.82 (0.60–1.12)	0.214	Fixed	0.0	0.686
Source of controls						
Clinical medical center	1	2.43 (1.12–5.26)	0.024	-	-	-
General population	3	0.92 (0.71–1.20)	0.551	Fixed	1.0	0.364

**Table 7 ijerph-18-02375-t007:** Subgroup analysis of the association between MEOHP and breast cancer risk.

Subgroups	No. of Studies	Meta-Analyses		Model	Heterogeneity	
		ORs (95% CIs)	*p*-Value		I^2^	*p*-Value
Region						
Non-America	2	0.92 (0.61–1.38)	0.686	Fixed	0.0	0.491
America	1	1.19 (0.71–1.99)	0.508	-	-	-
Source of controls						
Clinical medical center	1	1.15 (0.54–2.44)	0.716	-	-	-
General population	2	0.99 (0.70–1.40)	0.945	Fixed	0.0	0.333
All studies	3	1.01 (0.74–1.40)	0.927	Fixed	0.0	0.587

**Table 8 ijerph-18-02375-t008:** Subgroup analysis of the association between MCPP and breast cancer risk.

Subgroups	No. of Studies	Meta-Analyses		Model	Heterogeneity	
		ORs (95% CIs)	*p*-Value		I^2^	*p*-Value
Region						
Non-America	1	0.44 (0.24–0.80)	0.008	-	-	-
America	2	0.89 (0.63–1.25)	0.496	Random	70.9	0. 064
Source of controls						
Clinical medical center	1	1.02 (0.90–1.16)	0.760	-	-	-
General population	2	0.63 (0.46–0.85)	0.025	Fixed	43.9	0.182
All studies	3	0.74 (0.48–1.14)	0.173	Random	83.8	0.002

**Table 9 ijerph-18-02375-t009:** Subgroup analysis of the association between MBP and breast cancer risk.

Subgroups	No. of Studies	Meta-Analyses		Model	Heterogeneity	
		ORs (95% CIs)	*p*-Value		I^2^	*p*-Value
Region						
Non-America	2	0.77 (0.48–1.22)	0.266	Fixed	0.0	0.602
America	1	0.85 (0.47–1.64)	0.593	-	-	-
Source of controls						
Clinical medical center	1	0.66 (0.32–1.38)	0.267	-	-	-
General population	2	0.85 (0.56–1.30)	0.453	Fixed	0.0	1.000
All studies	3	0.80 (0.55–1.55)	0.228	Fixed	0.0	0.843

**Table 10 ijerph-18-02375-t010:** Subgroup analysis of the association between MiBP and breast cancer risk.

Subgroups	No. of Studies	Meta-Analyses		Model	Heterogeneity	
		ORs (95% CIs)	*p*-Value		I^2^	*p*-Value
Region						
Non-America	1	0.77 (0.48–1.22)	0.244	-	-	-
America	2	0.76 (0.57–1.03)	0.073	Fixed	0.0	0.740
All studies	3	0.75 (0.58–0.98)	0.033	Fixed	0.0	0.937

## Data Availability

No new data were created or analyzed in this study.

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
