# Peer review of "The Association of Bisphenol A and Phthalates with Risk of Breast Cancer: A Meta-Analysis"

_ijerph, 2021, doi:10.3390/ijerph18052375_

Round 1

Reviewer 1 Report

This meta-analysis of the correlation between breast cancer and BPA or various phtalates metabolite by Ge Liu et al. is well designed and described. This kind of work is needed in this field were somewhat conflicting reports are common. Besides the interesting associations between breast cancer and some phtalates metabolites, another key message is the paucity of good quality epidemiologic studies that the authors were able to identify.

A few suggestions to improve this manuscript:

1: The large majority of analysed studies are based on urine dosage of EDCs. Il would be interesting to the reader if the authors could discuss the limitations of these approaches. Among the potential caveats coming to mind, since several years often separate cancer initiation from its detection, exposure to EDC could no longer be detectable in patient's urine due to changes in exposure, e.g. following life style changes. In addition, could EDC detection in urine be altered by cancer or cancer treatments?

2: Albeit the English language is correct, proofreading by a native English speaker could further improve it.

3: In the penultimate paragraph of the Introduction, the authors mention an American study finding no association between phtalates or BPA and Breast cancer. I think the reference is missing.

3:

Author Response

Thank you for your comments concerning our manuscript entitled “The association of bisphenol A and phthalates with risk of breast cancer:A meta-analysis” (ID: 1116761). Those comments are all valuable and very helpful for revising and improving our paper, as well as the important guiding significance to our researches. We have studied comments carefully and have made correction which we hope meet with approval. Revised portion are marked in red in the paper. The main corrections in the paper and the responses are as follows:

Point 1: The large majority of analysed studies are based on urine dosage of EDCs. Il would be interesting to the reader if the authors could discuss the limitations of these approaches. Among the potential caveats coming to mind, since several years often separate cancer initiation from its detection, exposure to EDC could no longer be detectable in patient's urine due to changes in exposure, e.g. following life style changes. In addition, could EDC detection in urine be altered by cancer or cancer treatments?

Response 1: We appreciate very much your careful reading of our manuscript and professional suggestions. It is really true as you suggested that the limitation of conducting exposure assessment by urinary EDCs concentration should be discussed. As you mentioned, the use of EDCs in urine as a biomarker is influenced by some factors such as the relatively short biological half-life of EDCs, the changes in subjects' lifestyle after cancer diagnosis breast cancer itself and cancer treatments or surgery. We have considered the comments and have revised the manuscript accordingly. It may provide more ideas for readers in the field of the research.

Point 2: Albeit the English language is correct, proofreading by a native English speaker could further improve it

Response 2: Thanks for your valuable suggestions again. We have further polished our paper with a professional assistance in writing to make it more readable.

Point 3: In the penultimate paragraph of the Introduction, the authors mention an American study finding no association between phthalates or BPA and Breast cancer. I think the reference is missing.

Response 3: We are very sorry for our negligence of citing the reference. we have checked the manuscript and the corresponding reference is supplemented

We tried our best to improve the manuscript and made some changes in the manuscript.  These changes will not influence the content and framework of the paper. We greatly appreciate your warm work and hope that the revised manuscript is now suitable for publication.

Once again, thank you very much for your comments and suggestions.

Reviewer 2 Report

The article “The association of bisphenol A and phthalates with risk of breast cancer: a meta-analysis”, focuses on the association between BPA and phthalates and breast cancer by analyzing data from different studies. It is a relevant new study indicating that more high-quality case-control studies need to be done to clarify the effects of both toxic chemicals on th risk of developing breast cancer.

On the other hand, the methodology is clearly explained and correctly designed.

I only have 2 simple comments:

- a period is missing at th end of the abstract

- in the methods section, 2.1, the authors indicate that “irrelevant literatures” were excluded. What do they mean by that? I Think they should be more specific.

In summary, this manuscript is very interesting and shows nice and consistent data. I consider that the article is suitable for publication in “International Journal of Environmental Research and Public Health”, with the aforementioned considerations.

Author Response

Thank you for your comments concerning our manuscript entitled “The association of bisphenol A and phthalates with risk of breast cancer:A meta-analysis” (ID: 1116761). Those comments are all valuable and very helpful for revising and improving our paper, as well as the important guiding significance to our researches. We have studied comments carefully and have made correction which we hope meet with approval. Revised portion are marked in red in the paper. The main corrections in the paper and the responses are as follows:

Point 1: a period is missing at the end of the abstract

Response 1: Thanks for your careful comments on our paper. We are very sorry for our negligence of punctuation and we have already added a period at the end of the abstract.

Point 2: in the methods section, 2.1, the authors indicate that “irrelevant literatures” were excluded. What do they mean by that? I Think they should be more specific.

Response 2: “Irrelevant literatures” means some original research which have nothing to do with exploring the association of BPA and phthalates with human breast cancer risk in terms of research design, research objects, and research purpose, such as cell experiments, animal experiments, and the studies aimed on the association of other EDCs and other outcomes.

We tried our best to improve the manuscript and made some changes in the manuscript.  These changes will not influence the content and framework of the paper. We appreciate for your warm work earnestly, and hope that the correction will meet with approval.

Once again, thank you very much for your comments and suggestions.

This manuscript is a resubmission of an earlier submission. The following is a list of the peer review reports and author responses from that submission.